# Improved Wound Healing by Naringin Associated with MMP and the VEGF Pathway

**DOI:** 10.3390/molecules27051695

**Published:** 2022-03-04

**Authors:** Jia-Hau Yen, Wan-Ting Chio, Chia-Ju Chuang, Hsin-Ling Yang, Sheng-Teng Huang

**Affiliations:** 1Cancer Research Center for Traditional Chinese Medicine, Department of Medical Research, China Medical University Hospital, Taichung 40402, Taiwan; hugtrghu@yahoo.com.tw (J.-H.Y.); andless20@gmail.com (W.-T.C.); isseychaung@gmail.com (C.-J.C.); 2Institute of Nutrition, College of Biopharmaceutical and Food Sciences, China Medical University, Taichung 40402, Taiwan; hlyang@mail.cmu.edu.tw; 3Department of Chinese Medicine, China Medical University Hospital, Taichung 40402, Taiwan; 4School of Chinese Medicine, China Medical University, Taichung 40402, Taiwan; 5An-Nan Hospital, China Medical University, Tainan 40402, Taiwan

**Keywords:** naringin, wound healing, MMPs, VEGFs, VEGFRs

## Abstract

This study aims to investigate the wound-healing effectiveness of the phenolic compound, naringin, both in vitro and in vivo. Male mice were shaved on their dorsal skin under isoflurane, a biopsy punch was made in four symmetrical circular resection windows (6 mm) to induce a wound. These excision wounds were used to study the topical effects of naringin in terms of various biochemical, molecular, and histological parameters. We observed a significant recovery in the wound area. Increased levels of MMP-2, 9, 14, TIMP-2, VEGF-A, and VEGF-R1 were induced by naringin in the HaCaT cells. The time course experiments further revealed that levels of VEGF-A and B increased within 36 h; whereas levels of VEGF-C decreased. In line with this, VEGF-R3 levels, but not VEGF-R1 and 2 levels, increased soon after stimulation; although the increase subsided after 36 h. Additionally, naringin cream upregulated wound healing in vitro. The blockage of VEGF by Bevacizumab abolished the function of naringin cream on cell migration. Histological alterations in the wounded skin were restored by naringin cream, which accelerated wound healing via upregulated expression of growth factors (VEGF-A, B, and C and VEGF-R3), and thus increased MMP-2, 9, 14 expressions.

## 1. Introduction

The healing of a skin wound presents a complex process involving a cascade of cellular functions. Immediately after an injury, multiple biological functions are activated and respond in a synchronized manner. The wound repair process commonly leads to the substitution of the superficial epidermis, mucosa, or fetal skin, and skin repair displays an unspecific form of healing, wherein the wound is healed by a non-functional mass of fibrotic tissue known as scar tissue [1,2,3,4]. In fact, approximately 400 million people suffer from acute wounds annually, including traumatic wounds, burns, and surgical wounds [5]. Wounds can take longer to heal in patients with diabetes, resulting in increased risks of infection and of developing dangerous symptoms [6,7,8]. In addition, chronic wounds fail to heal by not following a predictable or expected healing pathway. The care of chronic wounds has thus evolved into its own specialty, with physicians often using a variety of therapies, including engineered skin, growth factors, pressure wound therapy (NPWT), and negative extracellular matrices (ECMs) [9].

Diminished production of vascular endothelial growth factor (VEGF) and decreased angiogenesis are thought to contribute to impaired tissue repair in diabetic patients [9,10]. Therefore, the potential use of therapeutic angiogenesis to improve wound healing has attracted great interest. Vascular endothelial growth factors (VEGFs) are considered powerful therapeutic tools for proangiogenic and pro-lymphangiogenic therapy for the treatment of several diseases [11]. The VEGF family of growth factors currently contains five other known members, including Placenta Growth Factor (PLGF), VEGFA, VEGFB, VEGFC, and VEGFD, which are key regulators of physiological and pathological vasculogenesis, angiogenesis, lymphangiogenesis, and vascular permeability [12,13,14,15]. These functional effects are controlled via the class III subfamily of tyrosine kinase receptors VEGFR1 (Flt-1), VEGFR2 (KDR/Flk-1), and VEGFR3 (Flt-4). Of these, VEGFR1 and VEGFR2 play major roles in angiogenesis, whereas VEGFR3 is primarily involved in lymph-angiogenesis [16]. Recent studies have reported that overexpression of VEGF-C via an adenoviral vector could improve the healing of full-thickness punch biopsy wounds in genetically diabetic (db/db) mice [17]. Indeed, when the function of endogenous VEGF-C/VEGF-D is blocked with a specific inhibitor, wound closure is delayed even further [17]. Phenolic compounds, including naringin, are characterized by suppressive effects on reactive oxygen species (ROS) generation and pro-inflammatory cytokines’ production [18,19,20,21,22,23]. Naringin is an active flavanone glycoside extracted from tomatoes, grapefruits, and related citrus fruits, which has been reported to exert protective effects against atherosclerosis development, with hepato-protective [24], cardio-protective [25], reno-protective [26], and neuro-protective effects [27]. Further investigations are necessary to validate the effectiveness of naringin in wound healing and inflammation. Thus, the present study aims to investigate the therapeutic value of naringin as a potential candidate to activate the VEGF pathway, thereby contributing to wound healing.

## 2. Results

### 2.1. Naringin Cream Improves Wound Healing in a Mouse Model

To investigate the effects of naringin cream on wound healing in vivo, we applied the naringin cream topically to full-thickness excisional wounds. Two different naringin doses were used (2.5 and 5 mg/wound). As shown in Figure 1, naringin treatment consistently accelerated wound closure, particularly at day 5, as compared with the control group. The naringin cream formulation also exhibited good homogeneity and extrudability. Our data suggest that naringin cream plays a positive role in the healing process of acute wound injuries.

### 2.2. Naringin Cream Treatment In Vivo

In Figure 2A, histological examination demonstrated that all groups treated with naringin showed significant closure as compared to the control and Gentermay treated wounds. The IHC results showed that on day 7, when wound healing was almost complete after naringin treatment, both the proteins of Ki67 and proliferating cell nuclear antigens (PCNA) had decreased expressions. The vessel number and CD31 level showed that the 5 mg naringin treatment group had the highest vessel numbers of the three treatment groups (Figure 2B). These results suggest that the enhanced wound healing provided by naringin cream treatment may be achieved via stimulating regeneration, including blood vessels. The expressions of VEGFA, VEGFB, and VEGFC were all increased in the skin wound tissues (Appendix A).

### 2.3. Upregulated Expressions of VEGFs and VEGFRs via Naringin Cream Treatment

To further assess the relevance of the results related to improve wound healing, we examined the levels of MMPs and the VEGF axis. HaCaT keratinocytes were used in the study, representing an in vitro model of proliferating and migrating keratinocytes. After exposure to the indicated dose of naringin, protein levels of MMP 2, 9 and 14, and VEGFA were increased; however, TIMP-2 and VEGF-R1 were only increased in response to 10 μM of naringin (Figure 3A,B). The level of TIMP-1 remained unchanged upon naringin treatment. Expressions of VEGFA and VEGFB were increased in a time-dependent manner at 36 h, but subsequently decreased at 48 h. In contrast, production of VEGFC increased soon after stimulation but subsided after 24 h, while VEGFD remained unchanged in response to naringin (Figure 4). As VEGF family production rates changed profoundly and dynamically, the levels of their receptor VEGFR3 also changed substantially.

### 2.4. Naringin Activates HaCaT Keratinocytes Migration

Aside from providing physical protection, wound dressings must interact with the wound and improve the healing process. To evaluate the effects of naringin on wound closure, human keratinocyte HaCaT cells were used. A cell proliferation assay (CCK-8) demonstrated that naringin did not affect HaCaT cell proliferation with 24 h treatment (Appendix A). As shown in Figure 5, the in vitro scratch-wound healing assay and migration assay showed that naringin significantly enhanced migration and the narrowing of the scratch area by 24 h. Considering the function of naringin on VEGFs, the anti-VEGF antibody bevacizumab (Avastin) was used to block the effect of naringin on VEGFs. After exposure to bevacizumab, the increased wound-healing and migration effects were lost. These results indicate that VEGFs play a critical role in the migration of upregulated keratinocytes induced by naringin.

## 3. Discussion

The acute skin wound-healing process may be summarized into four overlapping phases: hemostasis, inflammation, proliferation, and tissue remodeling [28]. In most mammals, the natural wound healing process, including scar formation and tissue fibrosis, is a highly evolved tissue-scale attempt to restore the critical barrier functions necessary for survival. There are several types of wounds, depending on a variety of factors including the source of the wound and any underlying issues that may lead to it. Abrasions, lacerations, punctures, burns, and avulsions are common types of skin injury. While multiple factors may impair wound healing, the factors influencing repair can generally be categorized into local and systemic factors [29]. Local factors are those which directly influence the characteristics of the wound itself, including oxygenation, infection, and venous sufficiency; while systemic factors involve the overall health or disease state of the individual which may impair the healing process [30]. Age and gender, sex hormones, nutrition, stress, ischemia, diabetes, keloids, fibrosis, jaundice, uremia, obesity, immunocompromised conditions (cancer, radiation therapy, AIDS), and other issues could be systemic factors [31,32]. In recent years, a growing body of research has been directed at understanding the critical factors influencing poor wound healing. Further clarification of the influences exerted on the repair process by these factors may contribute to the development of therapeutics to resolve impaired wounds. We evaluated the effects of naringin on different factors which affect cutaneous wound healing and the potential cellular and molecular mechanisms involved.

Naringin, a flavanone glycoside containing the flavanone naringenin and the disaccharide neohesperidose, has been one of the main active components in many Chinese herbal medicines for hundreds of years. It has been experimentally revealed to possess several biological properties such as anti-inflammatory, antioxidant, and anticancer activities. As shown in a recent study, naringin promotes osteoporotic fracture healing through increasing the VEGF level by interacting with its receptor VEGFR-2 [33]. The present study demonstrates the in vivo and in vitro effects of naringin, highlighting the potential pharmacological value of its activities. A previous study reported that naringin exerts a significant effect on bone repair, where naringin may mimic estrogen and suppress osteoclastogenesis by modulating OPG and RANKL expressions leading to increased bone mineral density (BMD) and bone strength, as well as inhibition of urinary calcium excretion [34,35]. To investigate potential effects beyond its positive functions on bone formation, we utilized PEO and naringin to formulate a naringin cream, thereby discovering that naringin cream could be greatly beneficial to wound healing. During treatment with the naringin cream, the regenerative response was coupled with efficient wound closure, consistent with the dynamic activation of the VEGFs’ pathway. In terms of VEGF-A and B production, we noted a clear increase in the initial 36 h, followed by a decline until 48 h; while VEGF-C increased only to 12 h. VEGF-D was not affected by naringin treatment. Of note, VEGFR3 expression was increased, but not that of VEGFR1 and 2, indicating that naringin treatment predominantly induces expression of VEGFA, B, C, and VEGFR3. During wound regeneration, the tissues display alternations in the expressions of MMPs and TIMPs, and both overproduction of MMPs or underproduction of its specific inhibitor TIMP-2 may result in matrix degradation and regeneration. Concurrently increased expressions of MMP-2 and MMP-9 were found within 12 h, after which point, they decreased. Meanwhile, the decrease of TIMP-2, as an MMP-2 and MMP-9 inhibitor, suggests the existence of a complex mutual interaction [36]. To explore the precise expression balance between MMPs and TIMPs, and thereby further clarify their roles in ECM regeneration during wound healing, we observed that TIMP-2 was decreased after naringin treatment at 12 h, and upregulated after 36 h. The blockage of VEGF-mediated MMP activation by bevacizumab abolished or perturbed the function of naringin in the wound healing assay and migration in vitro. These findings suggest that activation of the VEGF signaling pathway and relatively higher expressions of MMP-2 and 9 may contribute to the function of naringin serving as a prognostic initiator in wound healing. Additionally, we found that the wound contraction was almost completed in the 5 mg naringin-treated group at day 7. Thus, we choose day 7 to sacrifice the mice for obtaining wound tissue. At day 7, the wound healing process in the naringin-treated group was in the remodeling phase, we hypothesized that in this moment the cells had passed from an uncommitted proliferation toward a specialized cell fate. For that reason, we found higher CD31 but not PCNA and Ki67.

## 4. Materials and Methods

### 4.1. Animals and Cell Culture

Male C57BL/6 mice (20–25 g) were provided by the National Laboratory Animal Center (NLAC), NARLabs, Taiwan. The mice were housed under a 12 h day/night cycle with water and commercial pellet food ad libitum. Five mice were involved in each group including control (cream only), 2.5 mg naringin, 5 mg naringin and Gentermay with three independent experiments. All experiments were completed under the approval of the Institutional Animal Care and Use Committee at China Medical University (Taichung, Taiwan) (2019-188-1). Human keratinocyte cell line, HaCaT cells, were grown in DMEM supplemented with 10% FBS and antibiotics at 37 °C in a 5% CO_2_ incubator.

### 4.2. Preparation of Naringin Ointment

Naringin was purchased from Sigma Chemical Co (St. Louis, MO, USA). The naringin (5/2.5 mg) ointment was composed of 70% of PEG400 (SCRC, Shanghai, China) and 30% of PEG4000 (SCRC, Shanghai, China). The naringin, PEG400 and PEG4000 ingredients were mixed and stirred until well-distributed at 80 °C, and then stirred into a semi-solid state at room temperature. Gentermay, a product that contains gentamicin in a cream base, served as the positive control. Gentermay was used at 5 mg/wound. The cream was applied to the wounds once per day.

### 4.3. In Vivo Wound Model

The mice were shaved on their dorsal skin under anesthesia with 2% isoflurane carried in oxygen, and then a biopsy punch made four symmetrical circular resection windows (6 mm) to induce a wound. At days 1, 3, 5 and 7 past wounding, the wound regeneration was recorded by macroscopic images. At the end point of the experiment, the animals from each group were sacrificed with an overdose of isoflurane. For the histopathological analysis, the skin tissue was fixed in 4% formaldehyde solution in phosphate buffered saline (PBS) on ice for 10 min, dehydrated, and embedded in paraffin. The tissue paraffin was cut into 5 μm sections. Fixed sections were then stained with hematoxylin and eosin (HE). The sections were analyzed using an optical microscope (Nikon 80, Tokyo, Japan). Immunohistochemical analyses were performed using anti-PCNA (13110, Cell Signaling), Ki67 (ab16667, abcam) and CD31 (ab28364, abcam). The sections were incubated overnight with Abs at concentrations of 0.5–5 μg/mL at 4 °C. After incubation with peroxidase-conjugated secondary Abs, the chromogen diaminobenzidine tetrahydrochloride was added. The sections were imaged with a transmission microscope (Karl Zeiss Axio Observer Z1) at 200× magnification.

### 4.4. Histopathological Evaluation

Once the mice were sacrificed, the dorsal skins were fixed with 3.7% neutral buffered formalin at room temperature for 24 h. Seven-micrometer serial sections were clipped and stained with hematoxylin and eosin (HE), and examined for naringin ointment-induced histopathological changes that were scored as the area of wound and mouse skin epidermis. Skin thickness was measured using the Image J software (version 1.46; Madison, WI, USA, National Institutes of Health).

### 4.5. Western Blot

The HaCat cells cultured with or without naringin were harvested and total cell protein was extracted using whole cell lysis buffer. The protein concentrations were determined by the Bradford method (Bio-Rad, CA, USA). Samples with an equal amount of protein were subjected to 8–15% sodium dodecyl sulfate polyacrylamide gel electrophoresis (SDS-PAGE) and transferred onto a polyvinylidene difluoride (PVDF) (Millipore, Bedford, MA, USA) membrane. The membrane was incubated at room temperature in blocking solution (5% nonfat milk) for 1 h followed by incubation for 2 h in blocking solution containing an appropriate dilution of anti-MMP2 (ab86607, abcam), MMP-9 (ab76003, abcam), MMP-14 (ab51074, abcam), TIMP-1 (MAB3300, millipore), TIMP-2 (ab180630, abcam), VEGF-A (ab46154, abcam), VEGF-B(ab185696, abcam), VEGF-C (ab191274, abcam), VEGF-D (ab155288, abcam), VEGF-R1(ab32152, abcam), VEGF-R2 (9698, cell Signaling), VEGF-R3 (2485, cell Signaling), and β actin (E-AB-20058, Elabscience). After washing, blots were then probed with appropriate secondary horseradish peroxidase (HRP)-conjugated secondary antibodies (Jackson ImmunoResearch, West Grove, PA) and detected by an ECL detection system (Millipore). β-actin served as internal control.

### 4.6. Wound Healing Assay

In vitro migration was analyzed using Ibidi μ-Dish 35 mm (cat. No. 81176, Ibidi). HaCaT cells (70 μL; concentration: 5 × 10^5^ cells/mL) were added to a Culture-Insert well and cultured for 16 h and 24 h. The proliferation of cells was inhibited by treating with a proliferation inhibitor (actinomycin C). After removal of the Culture-Insert, cells were cultured for 16 and 24 h. The migration distance of cells was recorded and measured using Image J software.

### 4.7. Migration Assay

Cells were plated onto the Transwell^®^ Boyden chamber with or without naringin treatment on the upper chamber for migration assay. The chambers were incubated for 24 h with complete medium added in the lower chamber. Non-moved cells were removed by cotton swabs and the chambers were stained with crystal violet. Photomicrographs of three random regions were captured from duplicated assay chambers. The numbers of cells were counted and normalized to the untreated control.

### 4.8. Statistical Analysis

All statistical analyses were performed using GraphPad Prism statistical software (version 6, GraphPad Software, Inc., San Diego, CA, USA). Results were represented as means ± standard deviation (SD). One-way ANOVA was carried out when multiple comparisons were evaluated. Values were considered to be significant at *p* < 0.05. All experiments were repeated independently at least three times.

## 5. Conclusions

The current study demonstrates the effectiveness of naringin to accelerate and thus enhance the wound-healing process via several mechanisms. The in vivo wound-healing model revealed higher levels of CD31 detected in the mice used in this study. Interestingly, levels of PCNA and Ki67 in the naringin cream treated group were decreased compared to the control group. We hypothesize that this could be due to the times at which the analyses were performed after the puncture event. After seven days, regeneration was nearly complete, thus the cell proliferation markers PCNA and Ki67 were decreased in accordance with the wound size. However, CD31 is active in the later stages of the wound-healing process in order to rebuild the blood vessels, which could explain why we identified the dose-dependent increased level of CD31 via naringin. The exact mechanisms behind our observed improvement in wound healing by naringin treatment based on in vitro, ex vivo, and in vivo models could partially be explained by its combined ability to enhance cell migration, activate the VEGFs pathways, and upregulate the downstream effectors MMP-2 and 9 (Figure 6). We herein propose a possible association between VEGFs and the cell-modulatory properties of naringin, although additional studies are required to further elucidate the molecular mechanisms under different conditions, such as a burn or diabetes model.

## Figures and Tables

**Figure 1 molecules-27-01695-f001:**
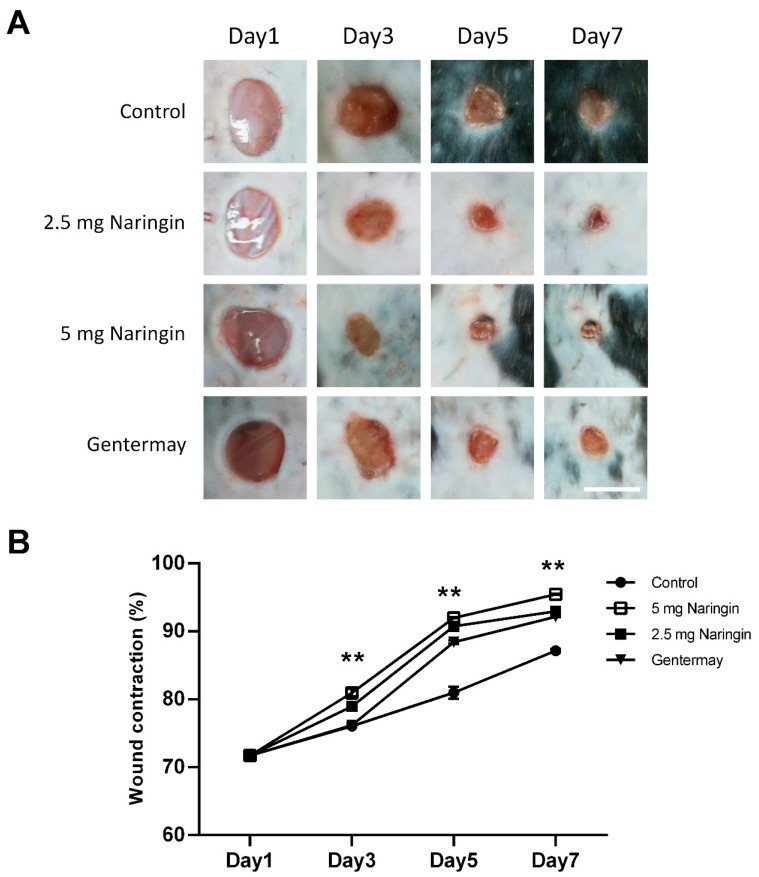
Naringin accelerates the closure of full-thickness punch biopsy wounds. (**A**) Morphological representation of mice wound showing various phases of wound healing; (**B**) Diagram of the kinetics of wound closure in mice treated with different conditions. Each point represents the mean percentage of regenerated wound size. Scale bar = 6 mm. Data are expressed as mean ± SEM and analyzed by one-way ANOVA analysis. n ≥ 5, ** *p* < 0.05 as compared with the vehicle control group.

**Figure 2 molecules-27-01695-f002:**
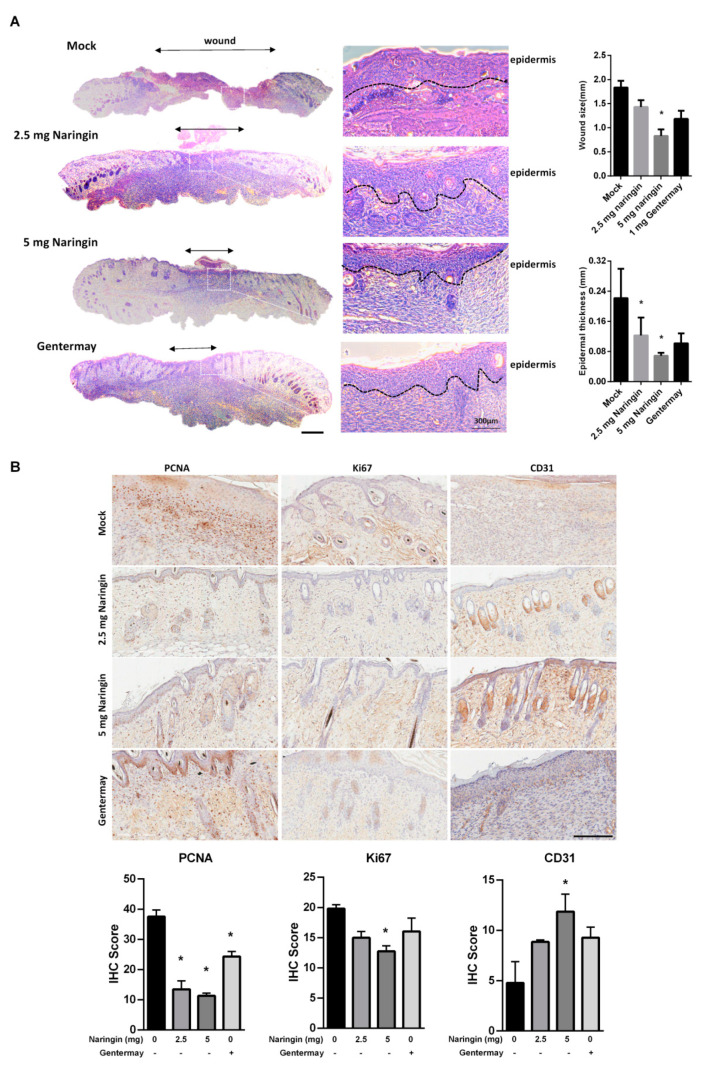
Naringin treatment showed better wound closure via examining dermal histopathological sections. (**A**) Representative HE staining images at day 7. Evaluation of granulation tissue area and thickness was noted. Scale bar = 1 mm; (**B**) IHC evaluation of the wound-healing quality. Wound sections were evaluated on day 7 by staining with anti-PCNA, Ki67 and CD31antibodies. Scale bar = 200 μm. Data are expressed as mean ± SEM and analyzed by one-way ANOVA analysis. n ≥ 5, * *p* < 0.05 as compared with the vehicle control group.

**Figure 3 molecules-27-01695-f003:**
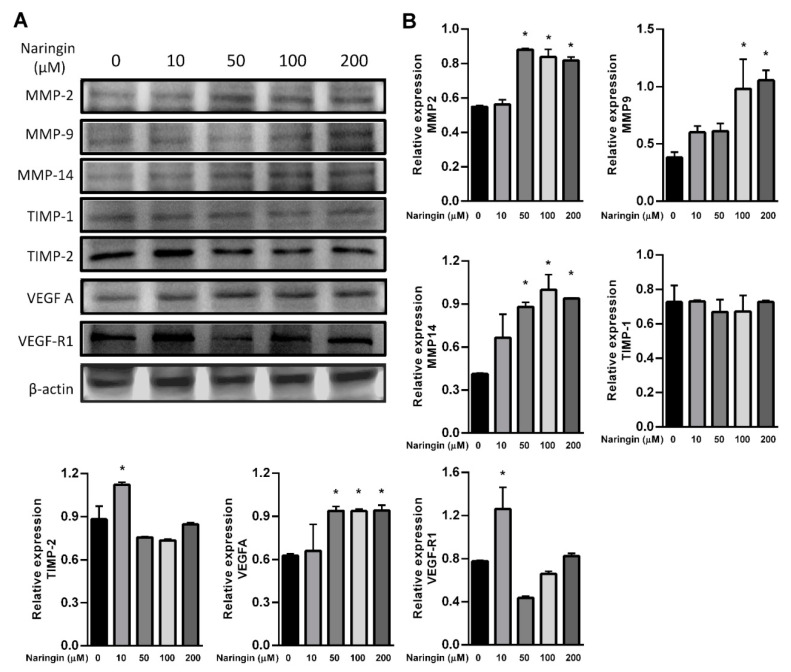
Naringin increases expression levels of VEGF-A, MMP-2, MMP-9 and MMP-14. (**A**) HaCaT cells were treated with or *without naringin* for 24 h and then the lysates were subjected to Western blot with specific antibodies; (**B**) Bar graphs show fold change compared to the intensity of β-actin in untreated samples from 3 independent experiments. Data are expressed as mean ± SEM and analyzed by one-way ANOVA analysis. * *p* < 0.05 as compared with the vehicle control group.

**Figure 4 molecules-27-01695-f004:**
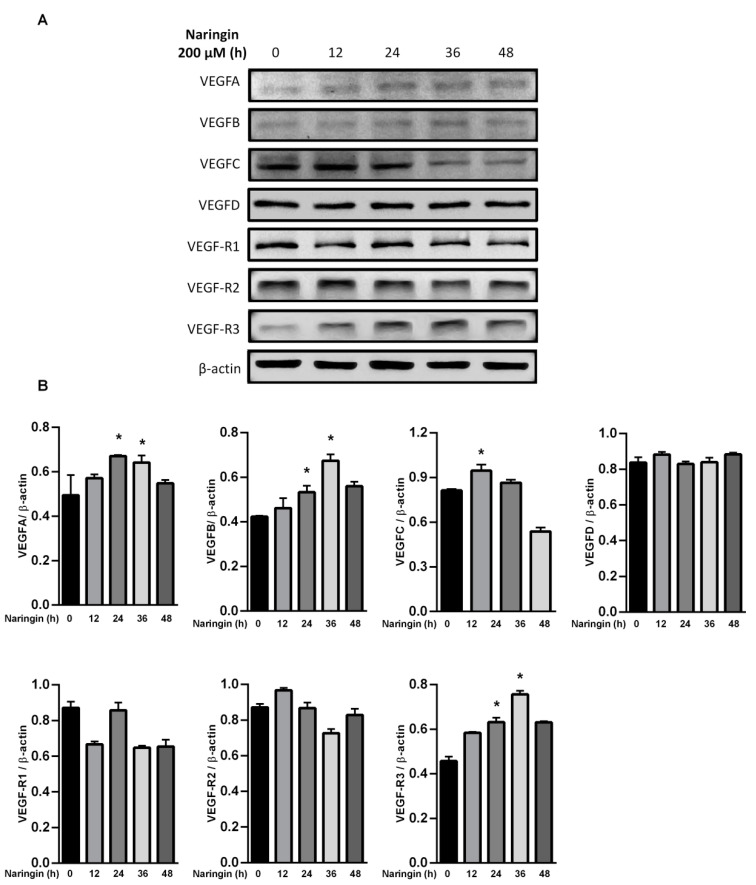
Naringin dynamically regulates VEGFA, B, and C. (**A**) HaCaT cells were treated with *naringin* for indicated times, and then the lysates were subjected to Western blot with specific antibodies; (**B**) Bar graphs show fold change compared to the intensity of β-actin in untreated samples from 3 independent experiments. Data are expressed as mean ± SEM and analyzed by one-way ANOVA analysis. * *p* < 0.05 as compared with the vehicle control group.

**Figure 5 molecules-27-01695-f005:**
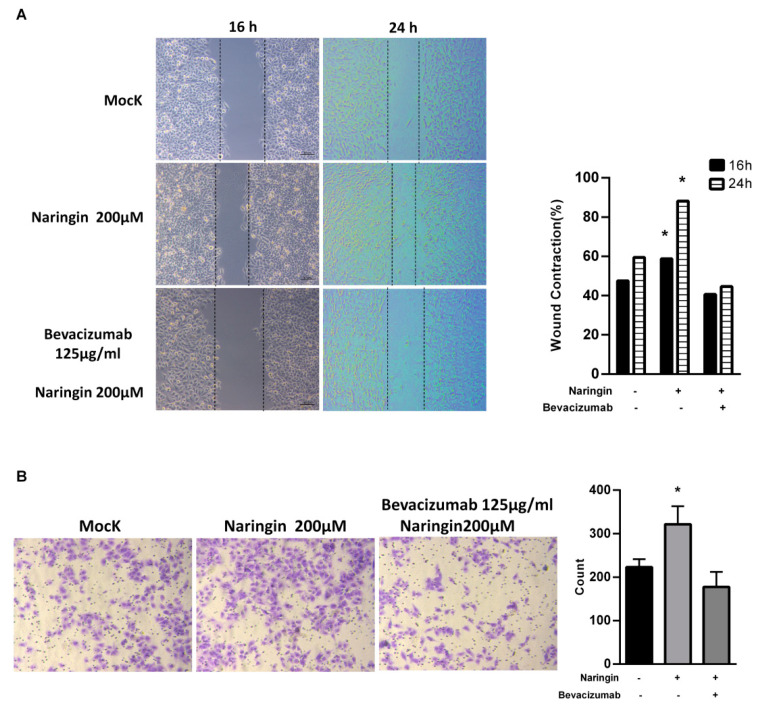
Naringin enhances migration of HaCaT cells. (**A**) Naringin-induced cells migration was tested by scratch healing assays. The images in 16 h and 24 h were analyzed for gap area over time; (**B**) Representative transwell migration from cells treated with different conditions. Bevacizumab functions as an inhibitor for VEGF. Data are expressed as mean ± SEM and analyzed by one-way ANOVA analysis. * *p* < 0.05 as compared with the vehicle control group.

**Figure 6 molecules-27-01695-f006:**
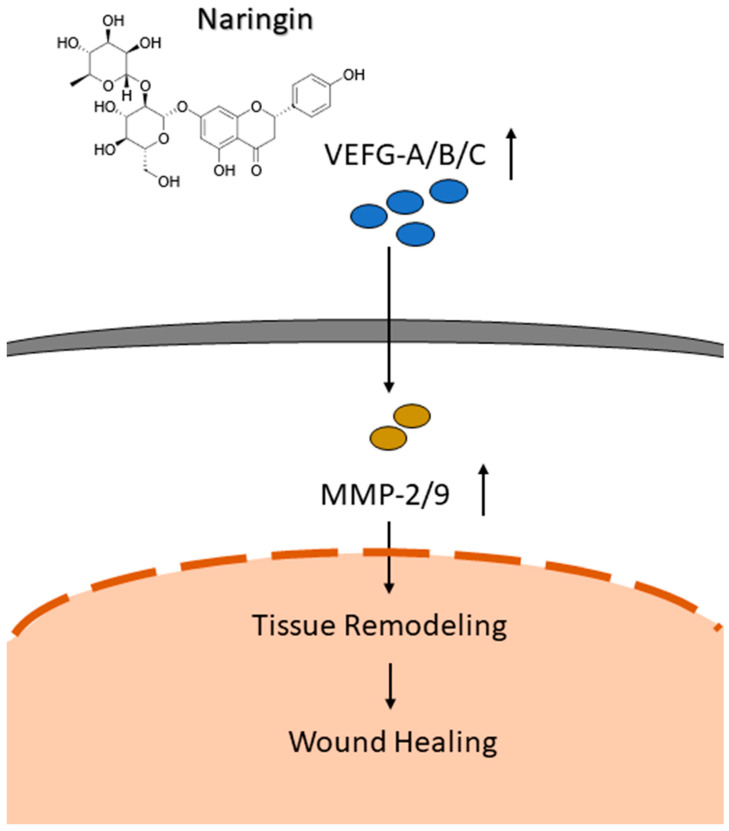
Schematic diagram of mechanism associated with naringin-induced cell migration to promote wound healing through activating the VEGFs pathways, and upregulating the downstream effectors MMP-2 and 9.

## Data Availability

The datasets used and/or analyzed during the current study are available from the corresponding author on reasonable request.

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
