# Peer review of "Improved Wound Healing by Naringin Associated with MMP and the VEGF Pathway"

_molecules, 2022, doi:10.3390/molecules27051695_

Round 1
Reviewer 1 Report
I wonder how such a pharmacology or in-depth activity study could be assigned to a group like natural product chemistry. Of course, Naringin is indeed a natural compound, but the whole article is about Naringin's wound-healing-related activity, which in my opinion doesn't have much to do with the natural product itself, uh, well. The authors explored various biochemical, molecular and histological changes of naringin in promoting wound healing, and proposed possible molecular mechanisms. It is a relatively complete and meaningful study that can be considered for publication in this journal. The following details need to be fixed or improved: 1-Line 202 CO2 2-line 49-71 Line spacing 3- The chemical structure of naringin is shown in Figure 6, whether it helps to improve the affinity of the article.Author Response
I wonder how such a pharmacology or in-depth activity study could be assigned to a group like natural product chemistry. Of course, Naringin is indeed a natural compound, but the whole article is about Naringin's wound-healing-related activity, which in my opinion doesn't have much to do with the natural product itself, uh, well. The authors explored various biochemical, molecular and histological changes of naringin in promoting wound healing, and proposed possible molecular mechanisms. It is a relatively complete and meaningful study that can be considered for publication in this journal. The following details need to be fixed or improved: 1-Line 202 CO2 2-line 49-71 Line spacing 3- The chemical structure of naringin is shown in Figure 6, whether it helps to improve the affinity of the article.
Response:
1) Thank you for your reminder. We have reformatted the word.
2) Thank you for your reminder. We have reformatted the part.
3) We have increased the chemical structure of naringin as shown in Figure 6.
Reviewer 2 Report
Please address the following:
1) Why did you choose naringenin for your study?
2) Why two doses of naringenin were studied?
3) Explain the mechanism in digramatic presentation more clearly.
Author Response
Comments and Suggestions for Authors
Please address the following:
- Why did you choose naringenin for your study?
Response: Drynaria fortunei J. Sm (D. fortunei), known as Gu-Sui-Bu, is used in traditional Chinese medicine for bone injury. We found that D. fortunei increased MMP-2 activity, thereby stimulating angiogenesis and cell migration, both in vivo and in vitro, as a result of MMP-2 and TIMP-2 balance modulation and the activation of VEGF/VEGFRs expression. Additionally, naringin is the major compound in D. fortunei. This is why we chose naringin as a candidate for wound healing in this study. These results have been published at Front Pharmacol. 2018 Sep 21;9:979. doi: 10.3389/fphar.2018.00979. eCollection 2018.
- Why two doses of naringenin were studied?
Response: We referred to Kandhare’s work published at Pharm Biol. 2016;54(3):419-32. A soft cream containing 1, 2, and 4% (w/w) naringin was formulated and evaluated for physicochemical characters. The naringin (5/2.5 mg) ointment composed of 70% of PEG400 and 30% of PEG4000 is approximately 5%, and 2.5% (w/w) naringin. Besides, we tried several times for wound healing to achieve the better results with these two doses.
- Explain the mechanism in digramatic presentation more clearly.
Response: Schematic diagram of mechanism associated with naringin-induced cell migration to promote wound healing through activating the VEGFs pathways, and upregulating the downstream effectors MMP-2 and 9.
Reviewer 3 Report
- There is no information about naringin-based cream formulation. It is a commercial cream? It was formulated by the authors? A detailed information about cream composition, its physicochemical properties and dosages must be included in the manuscript.
- Similarly, there is no information about the source of naringin. It was isolated by the authors? It was from a commercial source? There is some structural data and phytochemical characterization available?
- I suggest to include the chemical structure of naringin.
- The authors use some compound named gentermay…. but there is no information about that. This is a positive control? It is a commercially available compound? What is the formulation or active substance? Which dose was used?
- The in vivo experiments are not well described. The authors need to improve it. There is a different group of animals for each studied time-point? What are the number of animals per group? Please include the protocol number officially approved by the Institutional Ethics Committee and the date in which the approval was obtained.
- It is not clear if the samples analyzed in Fig. 5 are the same obtained from experiment presented in Fig. 1 or if they came from different experiments. I suggest to present and discuss all the in vivo data first and after move to the in vitro cell culture experiments.
- If there is one different group for each time-point, why the data for PCNA, Ki67 and CD31 were analyzed only at the day 7? How the authors explain a reduced labeling for PCNA and Ki67?
- Magnifications/scale must be included for each H&E and IHC section images. Why H&E sections on Fig 5A were edited? I suggest to present the whole section. I also suggest include some arrows indicating each layer or region that was used to take the measurements.
- Fig. 4 needs improvement. Quantifications of transwell migration experiment are bellow wound healing images, while quantification of wound healing are bellow transwell images. Letters indicating the panels A, B, C etc… are also missing.
- In Fig. 4 appear the commercial names Avasin and in the text Avastin. Please, use the name of active antibody bevacizumab in both text and figures. I also suggest to indicate the antibody concentration used per treatment and not only the amount in micrograms.
- The transwell experiment is not described in material and methods section.
- The authors must include some experiments to show the effects of naringin on HaCat proliferation and/or viability.
- In material and methods the authors mentioned that in vitro migration was analyzed using Ibidi culture inserts. Please include the catalog number of this material. There are some Ibidi plates that use a two-chamber silicone insert. When the silicone insert is removed it leaves an artificial gap between the chambers. This type of material estimates more accurately cell proliferation then migration, because it is not a scratch-induced wound. There is not a scratch-induced signaling stimulating the cells to migrate in this case. To estimate only cell migration using two-chamber silicone inserts the authors must included a proliferation blocker such as mitomycin C, for instance.
- Again in Fig. 4…. If the gap area formed by the inserts was analyzed over 16 h, why just one time-point is presented? In the case of transwell experiment what is the time-point presented? This information must be clear in the manuscript.
- The discussion section needs to be improved. In my opinion the basic information presented in lines 146 to 165 is not a priority in this case. The discussion needs to be focused on naringin and in the mechanisms showed by the authors in comparison to what have been published before. For instance, there are some papers describing angiogenesis promotion by naringin through VEGF/VEGFR2 pathway, but this is not discussed. The new data/mechanisms found by the authors needs to be highlighted.
- In the page 9, line 191, the authors mentioned “The blockage of MMP-9 by Avastin abolished/perturbed the function of naringin in the wound healing assay and migration in vitro”. In my opinion, this conclusion is not supported by the data. Bevacizumab targets specifically VEGF and not MMP-9. Bevacizumab can modulate VEGF-mediated MMP activation, but is not responsible for a direct MMP-9 inhibition. To properly address this issue the authors should include a MMP inhibitor in the experiments.
- Again, in page 9, Line 192-194, the authors mentioned: “These findings suggest that relatively higher expressions of MMP-2 and 9 against TIMP-2 may serve as a prognostic initiator in wound healing”. In my opinion this conclusion is also not supported by the data. Please explore more this discussion.

Author Response
In this work, Yen and colleagues investigated the mechanisms behind wound healing promotion induced by a naringin-based topical cream. Naringin is a well-known flavanone glycoside extracted from natural sources which is able to improve healing and angiogenesis related process. In this work, authors explore the effects of a topical naringin based formulation on wound healing, cell migration, inflammation and VEGF/VEGFR/MMP regulated pathways.
In general, the manuscript explores an interesting mechanism for this molecule. Some points that require revision or more attention by the authors are described below.
MAJOR and MINOR CONCERNS:
- There is no information about naringin-based cream formulation. It is a commercial cream? It was formulated by the authors? A detailed information about cream composition, its physicochemical properties and dosages must be included in the manuscript.
Response: We have added “4.2. Preparation of Naringin Ointment” in the section of Materials and Methods.
4.2. Preparation of Naringin Ointment
Naringin was purchased from Sigma Chemical Co (St. Louis, Mo, U.S.A.). The naringin (5/2.5 mg) ointment is composed of 70% of PEG400 (SCRC, shanghai, China) and 30% of PEG4000 (SCRC, shanghai, China). Mix the naringin, PEG400 and PEG4000 ingredients and stir until well-distributed at 80°C, and then stir into a semi-solid state at room temperature. Gentermay, contains gentamicin in a cream base, served as a positive control. 5 mg/wound Gentermay was used.
- Similarly, there is no information about the source of naringin. It was isolated by the authors? It was from a commercial source? There is some structural data and phytochemical characterization available?
Response: Thanks for your comments. Naringin was purchased from Sigma Chemical Co (St. Louis, Mo, U.S.A.). Additionally, the structure of naringin is also added in the Figure 6.
4.2. Preparation of Naringin Ointment
Naringin was purchased from Sigma Chemical Co (St. Louis, Mo, U.S.A.).
- I suggest to include the chemical structure of naringin.
Response: The structure of naringin has added in the Figure 6 by your suggestion.
- The authors use some compound named gentermay…. but there is no information about that. This is a positive control? It is a commercially available compound? What is the formulation or active substance? Which dose was used?
Response: We have added to describe gentermay in the section of Materials and Methods.
Gentermay, contains gentamicin in a cream base, is a wide spectrum antibiotic preparation for topical administration. 5 mg/wound Gentermay was used.
- The in vivo experiments are not well described. The authors need to improve it. There is a different group of animals for each studied time-point? What are the number of animals per group? Please include the protocol number officially approved by the Institutional Ethics Committee and the date in which the approval was obtained.
Response: Thanks for your reminder. We have improved the description in 4.1. In Vivo Wound Model as shown below.
Five mice were involved in each group including control (PEG cream only), 2.5 mg naringin, 5 mg naringin and gentermay with three independent experiments. All experiments were done under approval of the Institutional Animal Care and Use Committee at China Medical University (Taichung, Taiwan) (2019-188-1).
- It is not clear if the samples analyzed in Fig. 5 are the same obtained from experiment presented in Fig. 1 or if they came from different experiments. I suggest to present and discuss all the in vivo data first and after move to the in vitro cell culture experiments.
Response: We have corrected this part by your suggestion. Actually, they are the same experiments. We have changed the original Figure 5 to Figure 2 to make sense for in vivo data, and then discussed in vitro data. Thanks for your great suggestions.
- If there is one different group for each time-point, why the data for PCNA, Ki67 and CD31 were analyzed only at the day 7? How the authors explain a reduced labeling for PCNA and Ki67?
Response: We found the wound contraction was almost completed in the 5mg naringin-treated group at the day 7. Thus, we choose day 7 to sacrifice the mice to obtain wound tissue. At day 7, the wound healing process in naringin treated group were in remodeling phase so that we supposed that at this moment the cells passed from an uncommitted proliferating toward a specialized cell fate. That’s the reason why we found higher CD31 instead of PCNA and Ki67.
- Magnifications/scale must be included for each H&E and IHC section images. Why H&E sections on Fig 5A were edited? I suggest to present the whole section. I also suggest include some arrows indicating each layer or region that was used to take the measurements.
Response: We did not change the size of the results from H&E stain. In fact, the figures show equal scale of the tissues. Accordingly, we have modified the figures more clearly as shown in the Figure legend 1 and 2.
- Fig. 4 needs improvement. Quantifications of transwell migration experiment are bellow wound healing images, while quantification of wound healing are bellow transwell images. Letters indicating t also missing.
Response: We have corrected this part based on your comments. Thanks for your reminder.
- In Fig. 4 appear the commercial names Avasin and in the text Avastin. Please, use the name of active antibody bevacizumab in both text and figures. I also suggest to indicate the antibody concentration used per treatment and not only the amount in micrograms.
Response: We have corrected the commercial name of avastin to bevacizumab with its concentration instead of micrograms in Figure 5.
- The transwell experiment is not described in material and methods section.
Response: Thank you for your reminder. We described in the section of “4.7 . Migration assay” accordingly.
- The authors must include some experiments to show the effects of naringin on HaCat proliferation and/or viability.
Response: These data are now provided as shown in Supplementary Figure 2.
- In material and methods the authors mentioned that in vitro migration was analyzed using Ibidi culture inserts. Please include the catalog number of this material. There are some Ibidi plates that use a two-chamber silicone insert. When the silicone insert is removed it leaves an artificial gap between the chambers. This type of material estimates more accurately cell proliferation then migration, because it is not a scratch-induced wound. There is not a scratch-induced signaling stimulating the cells to migrate in this case. To estimate only cell migration using two-chamber silicone inserts the authors must included a proliferation blocker such as mitomycin C, for instance.
Response: These descriptions are now provided as new “4.6. Wound Healing Assay”. We also added the lost part of the assay as “The proliferation of cells was inhibited by treating the cells with a proliferation inhibitor (actinomycin C)”.
- Again in Fig. 4…. If the gap area formed by the inserts was analyzed over 16 h, why just one time-point is presented? In the case of transwell experiment what is the time-point presented? This information must be clear in the manuscript.
Response: The results of 16h and 24h were provided as new Figure 5A. Thank you for reminding us. We described the results from 24h as “4.7. Migration assay” accordingly.
- The discussion section needs to be improved. In my opinion the basic information presented in lines 146 to 165 is not a priority in this case. The discussion needs to be focused on naringin and in the mechanisms showed by the authors in comparison to what have been published before. For instance, there are some papers describing angiogenesis promotion by naringin through VEGF/VEGFR2 pathway, but this is not discussed. The new data/mechanisms found by the authors needs to be highlighted.
Response: We have increased the discussion based on your comments as shown below.
It has been experimentally revealed to possess several biological properties such as anti-inflammatory, antioxidant, and anticancer activities. As shown in a present study, naringin promotes osteoporotic fracture healing through increasing the VEGF level by interacting with its receptor VEGFR-2 [33].
- In the page 9, line 191, the authors mentioned “The blockage of MMP-9 by Avastin abolished/perturbed the function of naringin in the wound healing assay and migration in vitro”. In my opinion, this conclusion is not supported by the data. Bevacizumab targets specifically VEGF and not MMP-9. Bevacizumab can modulate VEGF-mediated MMP activation, but is not responsible for a direct MMP-9 inhibition. To properly address this issue the authors should include a MMP inhibitor in the experiments.
Response: Thanks for your kind reminder. The description was modified as shown below.
The blockage of VEGF-mediated MMP activation by Bevacizumab abolished/perturbed the function of naringin in the wound healing assay and migration in vitro.
- Again, in page 9, Line 192-194, the authors mentioned: “These findings suggest that relatively higher expressions of MMP-2 and 9 against TIMP-2 may serve as a
prognostic initiator in wound healing”. In my opinion this conclusion is also not supported by the data. Please explore more this discussion.
Response: Again, thanks for your kind reminder. We revised this part as shown below.
These findings suggest that activation of VEGF signaling pathway and relatively higher expressions of MMP-2 and 9 may contribute to the function of naringin serving as a prognostic initiator in wound healing.
Reviewer 4 Report
This study showed that naringin enhances wound healing in mouse skin wounds and scratch wounds in cultured HACAT cells. Naringin also increases the expression of VEGFs and VEGFRs as well as MMPs. Here are my comments and suggestions:
1: The paper did not give information about how to prepare naringin cream. Was the cream applied to wounds once or multiple times? What was the vesicle for the cream?
2: No description of Gentermay was provided.
3: Vesicle for the cream should be used as a negative control in in vivo study.
4: Fig1A what is the size of the scale bar?
5: Fig 3: Is VEGFC shown at the right in the middle panel supposed to be VEGFD?
6: Fig 4: 6h seems very short for cell migration. Please include 24 h data too.
7: Fig 5: a) what is IHC score? There was no description at all. b) Ki67 and PCNA should be examined at earlier time points such as day 1, 3, or 5 before the wounds were closed. c) Fig5B ki67 images shown were not in the wound bed areas. Its staining should be shown in areas shown in PCNA staining with epithelial cells. d) The authors did not describe cutaneous appendages at all in the results, but they concluded the treatment increased cutaneous appendages.
8: Fig 6 needs a legend.
9). VEGFc and VEGFR3 are essential for lymphangiogenesis during wound healing. The authors may want to examine lymphangiogenesis after the treatment.
10). Since the authors concluded that naringin promotes wound healing via VEGF pathway. To make the argument stronger, the author may want to check VEGF pathways in the skin wound tissues as done in cultured keratinocytes.
Author Response
Comments and Suggestions for Authors
This study showed that naringin enhances wound healing in mouse skin wounds and scratch wounds in cultured HACAT cells. Naringin also increases the expression of VEGFs and VEGFRs as well as MMPs. Here are my comments and suggestions:
1: The paper did not give information about how to prepare naringin cream. Was the cream applied to wounds once or multiple times? What was the vesicle for the cream?
Response: We have provided the description as shown in the section of 4.2. Preparation of Naringin Ointment.
4.2. Preparation of Naringin Ointment
Naringin was purchased from Sigma Chemical Co (St. Louis, Mo, U.S.A.). The naringin (5/2.5 mg) ointment is composed of 70% of PEG400 (SCRC, shanghai, China) and 30% of PEG4000 (SCRC, shanghai, China). Mix the naringin, PEG400 and PEG4000 ingredients and stir until well-distributed at 80°C, and then stir into a semi-solid state at room temperature. Gentermay, contains gentamicin in a cream base, served as a positive control. 5 mg/wound Gentermay was used. The cream was applied to the wounds once per day.
2: No description of Gentermay was provided.
Response: We have added to describe gentermay in the section of Materials and Methods.
Gentermay, contains gentamicin in a cream base, is a wide spectrum antibiotic preparation for topical administration. 5 mg/wound Gentermay was used.
3: Vesicle for the cream should be used as a negative control in in vivo study.
Response: Actually, the control is treated with PEG cream only (composed of 70% of PEG400 and 30% of PEG4000).
4: Fig1A what is the size of the scale bar?
Response: It’s 6mm. The lost note was added in the figure legend.
5: Fig 3: Is VEGFC shown at the right in the middle panel supposed to be VEGFD?
Response: Thank you for your reminder. We have corrected the mistakes.
6: Fig 4: 6h seems very short for cell migration. Please include 24 h data too.
Response: We have increased the data of 24h for cell migration as shown in Figure 5
7: Fig 5: a) what is IHC score? There was no description at all. b) Ki67 and PCNA should be examined at earlier time points such as day 1, 3, or 5 before the wounds were closed. c) Fig5B ki67 images shown were not in the wound bed areas. Its staining should be shown in areas shown in PCNA staining with epithelial cells. d) The authors did not describe cutaneous appendages at all in the results, but they concluded the treatment increased cutaneous appendages.
Response: a) We have added the part as 4.4. Histopathological Evaluation.
4.4. Histopathological Evaluation
Once the mice were sacrificed, the dorsal skins were fixed with 8% neutral buffered formalin at room temperature for 24 h. Seven-micrometer serial sections were clipped and stained with hematoxylin and eosin (H&E), and examined for naringin ointment-induced histopathological changes that were scored as the area of wound and mouse skin epidermis. Skin thickness was measured using the Image J software (version 1.46; Madison, WI, USA, National Institutes of Health).
- b) We sacrificed the mice at day 7 in our original design. Thus, we don’t have the tissue samples to examine Ki67 and PCNA at earlier time points including day 1, 3, or 5 before the wounds were closed.
- c) We have changed Figure 5B as shown in revised Figure 2B.
- d) The improper description has deleted.
8: Fig 6 needs a legend.
Response: Schematic diagram of mechanism associated with naringin-induced cell migration to promote wound healing through activating the VEGFs pathways, and upregulating the downstream effectors MMP-2 and 9.
9). VEGFc and VEGFR3 are essential for lymphangiogenesis during wound healing. The authors may want to examine lymphangiogenesis after the treatment.
Response: In fact, we are going to explore lymphangiogenesis in next preparation.
10). Since the authors concluded that naringin promotes wound healing via VEGF pathway. To make the argument stronger, the author may want to check VEGF pathways in the skin wound tissues as done in cultured keratinocytes.
Response: Thanks for your kind reminder. We have increased the data as shown in supplementary Figure 1. Our results demonstrated that the expressions of VEGFA, VEGFB and VEGFC were all increased in the skin wound tissues.
Round 2
Reviewer 3 Report
described bellow
Author Response
No comments from Reviewer #3. Thanks.
Reviewer 4 Report
The authors made some effort to improve the manuscript. Here are a few more my comments and suggestions:
1: Fig 2: Since PCNA and ki67 were downregulated in treated groups at day 7 and the data at an earlier time point such as day 1 or 3 are missing, I suggest the authors either add an explanation/limitations of the study design in the discussion or remove the data, but keep CD31 results which is more relevant to the current study.
2: Line 241: 3.7 % neutral buffered formalin is usually used for fixing tissues. Please double check the concentration.
Author Response
1: Fig 2: Since PCNA and ki67 were downregulated in treated groups at day 7 and the data at an earlier time point such as day 1 or 3 are missing, I suggest the authors either add an explanation/limitations of the study design in the discussion or remove the data, but keep CD31 results which is more relevant to the current study.
Response: We have added an explanation in the section of Discussion by your suggestion.
Additionally, we found the wound contraction was almost completed in the 5mg naringin-treated group at the day 7. Thus, we choose day 7 to sacrifice the mice for obtaining wound tissue. At day 7, the wound healing process in naringin treated group were in remodeling phase, we supposed that in this moment the cells passed from an uncommitted proliferating toward a specialized cell fate. That’s the reason why we found higher CD31 but not PCNA and Ki67.
2: Line 241: 3.7 % neutral buffered formalin is usually used for fixing tissues. Please double check the concentration.
Response: Thanks for your reminder. We have corrected.
Once the mice were sacrificed, the dorsal skins were fixed with 3.7% neutral buffered formalin at room temperature for 24 h.